# Top-GAP: Integrating Size Priors in CNNs for more Robustness, Interpretability, and Bias Mitigation

## Abstract

In the field of computer vision, convolutional neural networks (CNNs) have shown remarkable capabilities and are excelling in various tasks from image classification to semantic segmentation. However, their vulnerability to adversarial attacks remains a pressing issue that limits their use in safety-critical domains. In this paper, we present Top-GAP – a method that aims to increase the robustness of CNNs against simple PGD, FGSM, Square Attack and distribution shifts. The advantage of our approach is that it does not slow down the training or decrease the clean accuracy. Adversarial training instead requires many resources, which makes it hard to use in real-world applications. On CIFAR-10 with PGD $\epsilon = 8/255$ and 20 iterations, we achieve over 50% robust accuracy while retaining the original clean accuracy. Furthermore, we see increases of up to 6% accuracy against distribution shifts. Finally, our method provides the ability to incorporate prior human knowledge about object sizes into the network, which is particularly beneficial in biological and medical domains where the variance in object sizes is not dominated by perspective projections. Evaluations of the effective receptive field show that Top-GAP networks are able to focus their attention on class-relevant parts of the image.

## 1 Introduction

Modern computer vision has made remarkable progress with the proliferation of Deep Learning, particularly convolutional neural networks (CNNs). These networks have demonstrated unprecedented capabilities in tasks ranging from image classification to semantic segmentation (Zarándy et al., 2015). However, the robustness of these models remains a critical problem (DBL, 2018).

Many previous attempts to improve robustness have focused on adversarial training and additional (synthetic) images (Wang et al., 2023; Gowal et al., 2021). The disadvantage of these approaches is that both the computational complexity of the training drastically increases and the clean accuracy typically suffers (Clarysse et al., 2022; Raghunathan et al., 2019). A representative example is given by Peng et al. (2023), where standard adversarial training improves the robust accuracy on CIFAR-10 from 0% to 50.94% for ResNet-50, but the clean accuracy decreases from around 95% to 84.91%. Another example is Wang et al. (2023), where 50M training samples were generated, which inevitably leads to a strong increase in training time.

We propose a different approach that focuses on a novel method to regularize the network during training without adversarial samples. A constraint is added to the training procedure that limits the spatial size of the learned feature representation which a neural network can use for a prediction. Unlike Pathak et al. (2015), we do not need KKT conditions or the Lagrangian. The disadvantage of direct constrained optimization is that it can make gradient descent fail to converge if the algorithm is not modified. Instead, we force the network to only use the most important $k$ locations in the feature map. The "importance" stems from an additional sparsity loss that forces the network to output an empty feature map. Part of the loss tries to increase $k$ locations, while another part tries to set all of them to zero. This constraint simplifies the optimization problem and allows us to keep the same accuracy as the unconstrained problem.

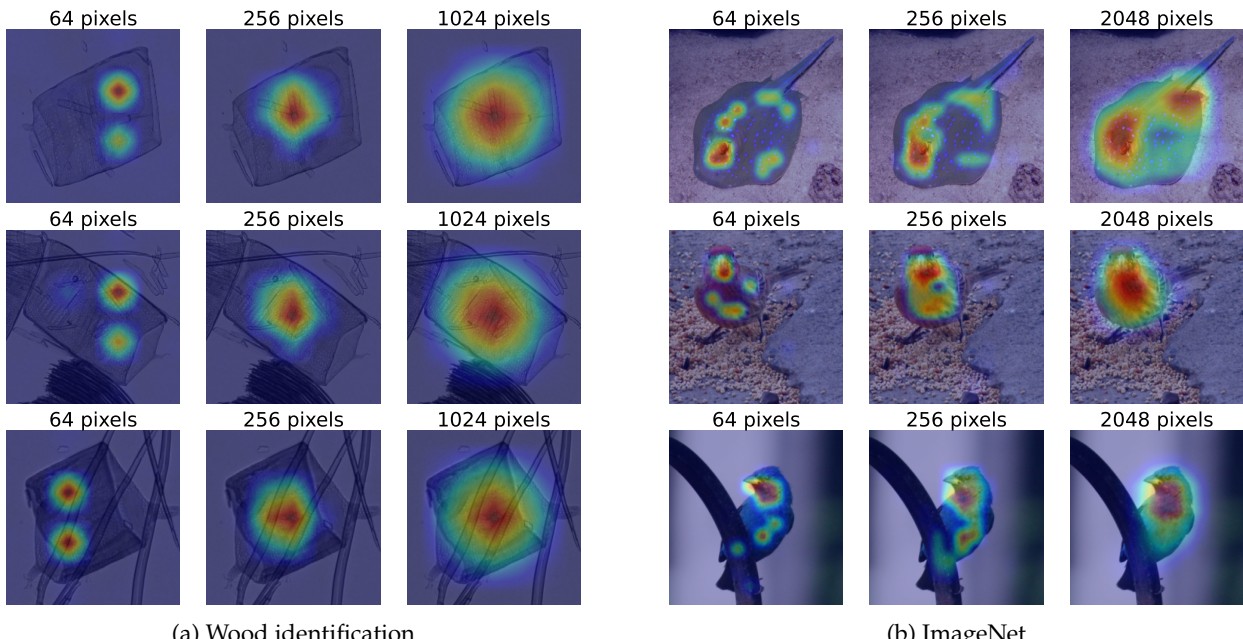

(a) Wood identification             (b) ImageNet

Figure 1: Example images from a biological classification dataset (a) and ImageNet (b), where we limit the number of pixels (i.e. locations in the output feature map) that the CNN can use to make predictions. Increasing the allowed pixel count leads to more pixels being highlighted in the class activation map (CAM). If the object size is not known or variable, the pixel constraint with the highest accuracy can be selected.

Restricting the output feature maps fundamentally changes the way the network works internally. In fig. 1, we see an example on how the constraint also affects the class activation map (CAM). We found that the networks trained with our approach become more robust. The intuition behind our proposed method is based on the observation that if the sample size of a class is too small, the network may tend to focus on the background instead of the object itself (Sagawa et al., 2019; Ribeiro et al., 2016). This can lead to undesirable biases in the classifier. In our approach, the constraint forces the network to not focus so much on the background.

The main contributions of this paper are:

- We introduce Top-GAP, a new approach to regularize networks by including a size prior in the network, which does not rely on the Lagrangian. This prior allows us to limit the number of pixels the network can use during inference. We show that this is especially beneficial for object classification tasks in imaging setups without perspective projections, such as biomedical imaging or benchmark datasets with centered objects.

- Extensive experiments on various architectures and datasets show, that our method improves robustness against certain cheap adversarial attacks while maintaining high clean accuracy. For attacks such as FGSM/PGD, we achieve an increase in native robustness of over 50% accuracy without adversarial training.

- Further, our evaluation shows, that even in the case of imaging setups with strong object size variations, we can still find a size constraint leading to improvements over baseline settings.

- Analyzing the effective receptive, we show that Top-GAP is able to steer the network to focus on object-pixels instead of the background.

- Finally, we report strong indications that our approach has the potential to mitigate bias. For example, when we take distribution shifts into account, we can achieve improvements in accuracy of up to 5%.

## 2   Related Work

Our work is related to different strands of research, each dealing with different aspects of improving the features and robustness of neural networks. This section outlines these research directions and introduces their relevance to our novel approach.

**Adversarial robustness.** It has been shown that neural networks are susceptible to small adversarial perturbations of the image (Goodfellow et al., 2015). For this reason, many methods have been developed to defend against such attacks. Some methods use additional synthetic data to improve robustness (Wang et al., 2023; Gowal et al., 2021). Wang et al. (2023) makes use of diffusion models, while Gowal et al. (2021) uses an external dataset. Other methods have shown that architectural decisions can influence robustness (Peng et al., 2023; Huang et al., 2022). For example, the Transformer-style patchify stem is less robust than a classical convolutional stem. A disadvantage of all these approaches is that the clean accuracy and training speed are negatively affected (Raghunathan et al., 2019; Clarysse et al., 2022). "Native robustness" (Grabinski et al., 2022) on the other hand, is defined in literature as robustness which is achieved by architectural changes only. The problem with regular adversarial training is also that the networks are usually only robust against a single perturbation type Tramèr & Boneh (2019). Hence, methods not using adversarial training are less expensive and less prone to overfitting on specific attacks.

**Bias mitigation and guided attention.** A notable line of research concentrates on channeling the network's focus towards specific feature subsets. Of concern is the prevalence of biases within classifiers, arising due to training on imbalanced data that perpetuates stereotypes (Buolamwini & Gebru, 2018). Biases may also stem from an insufficient number of samples (Burns et al., 2019; Zhao et al., 2017; Bolukbasi et al., 2016), causing the network to emphasize incorrect features or leading to problematic associations. For instance, when the ground truth class is "boat", the network might focus on waves instead of the intended object.

He et al. (2023); Yang et al. (2019) introduce training strategies to use CAMs as labels and refine the classifier's attention toward specific regions. In contrast, Rajabi et al. (2022) proposes transforming the input images to mitigate biases tied to protected attributes like gender. Moreover, Li & Xu (2021) suggests a method to uncover latent biases within image datasets.

**Weakly-supervised semantic segmentation (WSSS).** (Li et al., 2018) focuses on accurate object segmentation given class labels. The Puzzle-CAM paper (Jo & Yu, 2021) introduces a novel training approach, which divides the image into tiles, enabling the network to concentrate on various segments of the object, enhancing segmentation performance. There are many more publications that focus on improving WSSS (Sun et al., 2023). Some making use of foundational models such as Segment Anything Model (SAM) (Kirillov et al., 2023) or using multi-modal models like CLIP (Radford et al., 2021).

**Priors.** Prior knowledge is an important aspect for improving neural network predictions. For example, YOLOv2 (Redmon & Farhadi, 2016) calculated the average width and height of bounding boxes on the dataset and forced the network to use these boxes as anchors. However, there are many other works that have tried to use some prior information to improve predictions (Zhou et al., 2019; Cai et al., 2020; Hou et al., 2021; Wang & Siddiqi, 2016; Pathak et al., 2015). In particular, Pathak et al. (2015) has proposed to add constraints during the training of the network. For example, they propose a background constraint to limit the number of non-object pixels. However, they only train the coarse output heat maps with convex-constrained optimization. The problem is that the use of constraints can make it harder to find the global optima. Therefore, it is harder to train the whole network.

**Our approach.** Much like bias mitigation strategies and attention-guided techniques, we direct the network's focus to specific areas. However, our approach does not require segmentation labels and only minimally changes the CNN architectures. The objective is to maintain comparable clean accuracy and the number of parameters, while significantly improving the robustness and localization of objects. In contrast to WSSS, we do not intend to segment entire objects, but instead continue to concentrate on the most discriminative features. Given that we modify the classification network itself, we also diverge from methods that solely attempt to enhance CAMs of pretrained models.

## 3 Method

In most cases of image classification, the majority of pixels are not important for the prediction. Usually, only a small object in the image determines the class. Our approach is geared towards these cases. In contrast, many modern CNNs implicitly operate under the assumption that every pixel in an image can be relevant for identifying the class. This perspective becomes evident when considering the global average pooling (GAP) layer (Lin et al., 2014) used in modern CNNs. The aim of the GAP layer is to eliminate the width and height dimensions of the last feature matrix, thereby making it possible to apply a linear decision layer.

The GAP layer averages all locations within the last feature matrix without making a distinction between the positions or values. This means that a corner position is treated in the same way as a center position. We also note that each of the locations in the last feature matrix corresponds to multiple pixels in the input image. This is known as the receptive field. Now, we want to define more formally the terminology.

**Definition 3.1** (Effective receptive field). Let $X^{(p)}_{i^{(p)},j^{(p)}}$ be the feature matrix on the $p$th layer for $1 \leq p \leq n$ with coordinates $(i^{(p)}, j^{(p)})$. The input to the neural network is at $p = 1$ and the output feature map at $p = n$. Then the effective receptive field (ERF) of the output location $(i^{(n)}, j^{(n)})$ with respect to the input pixel $(i^{(1)}, j^{(1)})$ is given by $\frac{\partial X^{(n)}_{i^{(n)},j^{(n)}}}{\partial X^{(1)}_{i^{(1)},j^{(1)}}}$ (Luo et al., 2017). This definition assumes that each layer has only a single channel. For multiple output channels, we compute $\sum_{k=1}^{c^{(n)}} \frac{\partial X^{(n)}_{i^{(n)},j^{(n)},k}}{\partial X^{(1)}_{i^{(1)},j^{(1)}}}$ where $c^{(n)}$ are the channels of the last feature map. The ERF characterizes the impact of some input pixel on the output.

**Definition 3.2** (Global Average Pooling). The feature output of the neural network $X^{(n)}$ is averaged to obtain a single value. This operation is known as Global Average Pooling (GAP) and is defined as:

$$\text{GAP}(X^{(n)}) = \frac{1}{h^{(n)}w^{(n)}} \sum_{i=1}^{h^{(n)}} \sum_{j=1}^{w^{(n)}} X^{(n)}_{i,j},$$

where $h^{(n)}$ is the height and $w^{(n)}$ is the width of the output feature map. In practice, there is not only one channel but $c^{(n)}$ channels.

An example shall explain the two terms. In case of EfficientNet-B0 (Tan & Le, 2020), $X^{(n)}$ has dimension $7 \times 7 \times 1280$ for an input image of size $224 \times 224$ where $c^{(n)} = 1280$ are the channels. The GAP$(\cdot)$ operation reduces $X^{(n)}$ to a vector of size $1280 \times 1$. All of the $7 \times 7$ locations have an effect on the classification. With the help of the ERF, we can measure how much the $224^2$ input pixels contribute to the $7^2$ output locations.

Another method to analyze what the neural network focuses on are the so-called class activation maps. These methods modify $X^{(n)}$ so that we get a visualization of what is important for the neural network.

**Definition 3.3** (Class Activation Map). The product of multiplying the output tensor $X^{(n)}$ by some weight coefficient $W$ is known as a class activation map (CAM) (Zhou et al., 2015). The standard CAM, also known as "CAM", uses the weights of the linear decision layer $L$.

In the previous example, the linear decision layer $L$ would map the 1280 channels to $c^{(n+1)}$ class channels. The output of the CAM would be in this case $7 \times 7 \times c^{(n+1)}$. Each of the $c^{(n+1)}$ maps can be upsampled to obtain a visualization.

**Definition 3.4** (GradCAM). GradCAM is a generalization of CAM to non-fully convolutional neural networks (non-FCN) such as VGG. It is equivalent to the standard CAM for FCN like ResNet. It is defined as follows

$$\text{GradCAM}(X, c) = \text{ReLU}\left(\sum_k W_{k,c} X_k\right),$$

with $W_{k,c} = \text{GAP}\left(\frac{\partial L(X)_c}{\partial X_k}\right)$, $k$ being the channel index of $X$ and $c$ being the index of the linear layer. Usually the last feature map $X^{(n)}$ is chosen for $X$.

In addition to GradCAM, there are many other CAM methods. However, they are all based on reducing the channels of $X^{(n)}$ in order to obtain a visualization. Instead of improving GradCAM, as so many approaches have done before (Chattopadhay et al., 2018; Omeiza et al., 2019; Jiang et al., 2021; Fu et al., 2020; Wang et al., 2020), we propose that the output of the CNN should be both a CAM and a prediction. Then we can regularize the CAM during training and can more fundamentally influence what is highlighted in the CAM.

Our approach involves integrating an object size constraint directly into the network, designed to enforce the utilization of a limited set of pixels for classification. This constraint allows for noise reduction and the elimination of unnecessary pixels from the CAM. In cases where specific-sized features determine the class, we can incorporate this prior knowledge into the neural network, enhancing its classification accuracy.

Before introducing the object constraint, we first change the model structure to output a higher-resolution CAM.

### 3.1 Changing the model output structure

Figure 2 shows the general structure of our architecture. The backbone can be any standard CNN such as VGG (Simonyan & Zisserman, 2015), ResNet (He et al., 2015), ConvNeXt (Liu et al., 2022) or EfficientNet (Tan & Le, 2020). Depending on the backbone, we use the last 3 or 4 feature maps as input to a feature pyramid network (FPN) (Lin et al., 2017). We note that the original FPN as used for object detection was simplified in order to reduce parameters. All the feature maps are upsampled to the size of the largest feature map and added together. We found no advantage in using concatenation. This output is given to a final convolutional layer that has the number of output classes as filters. Note that a convolutional layer with kernel size 1 is used for the implementation of the final linear layer. Optionally, dropout can be applied as regularization during training. Lastly, a pooling layer such as GAP is employed to obtain a single probability for each class.

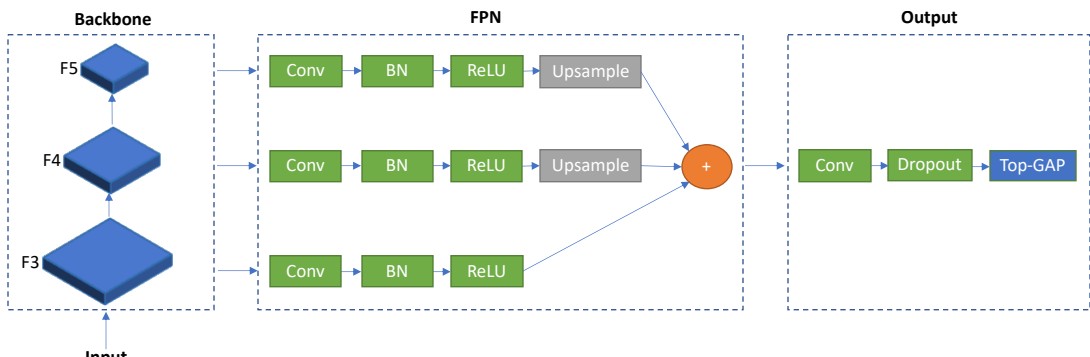

Figure 2: Example of our architecture applied to a backbone with 3 feature maps (e.g. $7 \times 7$, $14 \times 14$, $28 \times 28$). For all convolutions except the final one, a kernel size of $3$ and $256$ filters is used. The last convolution employs a kernel size of $1$, with the number of filters set to match the number of output classes. The CAM is as large as the biggest feature map (here F3). Our pooling layer ("Top-GAP") averages the CAMs given by the last convolutional layer ("Conv") to create a vector containing the probability for each class. For the CAM, we disable "Top-GAP" and perform min-max scaling.

For convenience, we explicitly define two modes for our model (refer to fig. 2):

  1. training/prediction: Top-GAP is enabled to obtain the probabilities for each class.

2. CAM: Top-GAP is disabled. The output feature map is upsampled to the size of the input image and normalized to be in the range $[0, 1]$.

Without our modified model, we would need to use a method such as GradCAM to obtain a visualization.

Let us compare the two approaches: EfficientNet-B0 with GradCAM and EfficientNet-B0 with our output structure (see fig. 2). GradCAM does not require any additional parameters because it generates the activation map from the model itself. If we change the model structure, we have more parameters, but also more influence on what is seen in the CAM. If we were to replace GradCAM with LayerCAM or some other method, it would never have the same impact as changing the model training itself (our approach). In addition, GradCAM does not combine multiple feature maps by default to achieve better localization.

In our approach, the standard output linear layer of some classification model like EfficientNet-B0 is substituted with $f + 1$ convolutional layers, where $f$ corresponds to the number of feature maps (refer to fig. 2). This leads to a small increase in the number of parameters.

| Architecture | Params (unmodified) | Params (ours) |
|---|---|---|
| VGG11-BN | 132.87M | 12.43M |
| EfficientNet-B0 | 4.08M | 4.75M |
| DenseNet-121 | 7.98M | 8.03M |

Table 1: Number of parameters for some architectures. We have less parameters than VGG because all additional linear layers are removed.

As indicated in table 1, we can achieve a comparable number of parameters.

These changes to the model are prerequisites for enabling the integration of size constraints within the neural network. If only the last feature map were used, a single value would correspond to an excessively large area in the original image. Hence, combining multiple feature maps proves advantageous. This idea is reinforced by findings from Jiang et al. (2021), which highlight that employing multiple layers enhances the localization capabilities of CAMs.

### 3.2 Defining the pixel constraint (Top-GAP)

Instead of using the standard GAP layer, we replace the average pooling by a top-k pooling, where only the $k$ highest values of the feature matrix are considered for averaging. This pooling layer limits the number of input pixels that the network can use for generating predictions.

In a standard CNN, the last feature map is at layer $n$. In our model (fig. 2), the last feature map is at $n + 1$ because we replaced the linear decision layer $L$ by a $1 \times 1$ convolution.

**Definition 3.5** (Top-GAP). We define the Top-GAP layer as follows:

$$\text{Top-GAP}(\tilde{X}, k)_t = \frac{1}{k} \sum_{i=1}^{k} \tilde{X}_{i,t} \,,$$

where $\tilde{X}$ represents the ordered feature matrix $X^{(n+1)}$ with dimensions $h^{(n+1)} w^{(n+1)} \times c^{(n+1)}$, where $c^{(n+1)}$ corresponds to the number of output classes. Each of the $c^{(n+1)}$ column vectors is arranged in descending order by value, and $k$ values are selected. $i$ indicates the ranking, with $i = 1$ being the largest value and $i = k$ being the smallest. $t$ is an index indicating the channel. We select for each channel different values.

When $k = 1$, we obtain global max pooling (GMP). When $k = h^{(n+1)} w^{(n+1)}$, the layer returns to standard GAP. The parameter $k$ enforces the pixel constraint, and its value depends on the image size. For instance, if the largest feature map has dimensions $56 \times 56$, then $\frac{k}{56^2}$ values are selected. Hence, when adjusting this parameter, it is crucial to consider the relative object size in the highest feature map.

### 3.3 Classification loss function

The last component of our method involves changing the loss function. While the Top-GAP$(\cdot)$ layer considers only locations with the highest values, these locations might not necessarily be the most important ones. Thus, it becomes essential to incentivize the reduction of less important positions to zero.

To achieve this, we add an $\ell_1$ regularization term to the loss function, inducing sparsity in the output. The updated loss function is defined as follows:

$$L = \lambda ||X^{(n+1)}||_1 + \text{CE}\left(\hat{y}, y\right) , \tag{1}$$

where $\text{CE}(\hat{y}, y)$ represents the cross-entropy loss between the prediction $\hat{y} = \text{softmax}\left(\text{Top-GAP}(\tilde{X}, k)\right)$ and the ground truth $y$. $\tilde{X}$ is the ordered $X^{(n+1)}$ feature output in our model, while $k$ is a fixed non-trainable parameter. Here, $\lambda$ controls the strength of the regularization. We found that for most datasets $\lambda = 1$ is sufficient.

The main difference between the regular classification loss and our loss is the addition of top-k pooling in conjunction with $\ell_1$ regularization.

Lastly, it is important to highlight that it is the combination of these distinct components that yields good results. In our ablation study, we will demonstrate that removing specific components lead to either reduced accuracy or worse feature representations.

## 4 Evaluation

In this section, we will systematically verify the claims of our method on multiple datasets. We give a detailed description of the datasets in appendix A. These datasets were chosen to have varying characteristics (different class counts, domains and object sizes). We test for each dataset multiple architectures.

We train all models except ImageNet using stratified cross-validation. The results obtained from each fold are then averaged. Employing cross-validation mitigates the impact of randomness on our findings (Picard, 2021).

The main hyperparameter of our approach is given by the pooling layer Top-GAP$(\cdot, k)$. This layer defines the constraint. We always combine this layer with our model structure (see fig. 2) and $\ell_1$ loss. We test for all the numerical experiments the values $k \in \{64, 128, 256, 512, 1024, 2048\}$, except for CIFAR-10 where we test $k \in \{8, 16, 32, 64, 128, 256\}$. Due to the high computational cost of training on ImageNet, ResNet-18 was only trained on $k = 256$.

For an image of size $224 \times 224$, the output CAM has dimensions $56 \times 56$. This means that we use approx. $\frac{64}{56^2} \approx 2\%$ of the feature map for $k = 64$. The highest value that we tested corresponds to $\frac{2048}{56^2} \approx 65\%$.

### 4.1 Hypothesis: certain input pixels become less important for classification

We want to show that with our method not all pixels in the input image have the same influence on the output feature map $X^{(n+1)}$.

In most datasets (e.g. ImageNet), the object to be classified is located in the center of the input image. While each pixel in the input image corresponds to multiple values in the output feature map $X^{(n+1)}$, the general position is the same. The center in the output is also the center in the input.

The input pixels should contribute much more to the center than to the background of $X^{(n+1)}$. We want to quantify how much influence the input pixels have on the center of $X^{(n+1)}$ and on the corner of $X^{(n+1)}$. For this, we use definition 3.1 and define a metric.

**Definition 4.1** (ERF distance). We define $\text{ERF}(1,1) = \frac{1}{hw}\sum_{i,j}\left|\frac{\partial X_{1,1}^{(n+1)}}{\partial X_{i,j}^{(1)}}\right|$ to be the absolute change of the output corner position $(1,1)$ with all input pixels $(i,j)$. Similarly, we define $\text{ERF}(\frac{h}{2}, \frac{w}{2})$ to be the change of

the output center position with respect to the input, where $h$ and $w$ is the width of the output feature map. Then the ERF distance is $\text{ERF}(\frac{h}{2}, \frac{w}{2}) - \text{ERF}(1, 1)$.

Intuitively, we expect a low value for $\text{ERF}(1, 1)$ because the corner position of the feature map contains less information. Similarly, $\text{ERF}(\frac{h}{2}, \frac{w}{2})$ should be a high value because the object is in the center. If the difference between the two values is low, it means that each pixel contributes similarly to the output.

| Dataset | Arch | ERF distance ↑ | ERF distance (ours) ↑ |
|---|---|---|---|
| COCO (Lin et al., 2015) | EN | 0.108 | **0.447** |
| COCO (Lin et al., 2015) | CN | 0.072 | **0.288** |
| COCO (Lin et al., 2015) | RN | 0.273 | **0.399** |
| Oxford (Parkhi et al., 2012) | EN | 0.013 | **0.383** |
| Oxford (Parkhi et al., 2012) | RN | 0.060 | **0.443** |
| CUB-200-2011 (Wah et al., 2011) | EN | -0.033 | **0.480** |
| CUB-200-2011 (Wah et al., 2011) | CN | -0.034 | **0.242** |
| CUB-200-2011 (Wah et al., 2011) | RN | 0.092 | **0.529** |

Table 2: The table shows that our approach leads to a different ERF. The center has a different effect than the corner of the image. "Ours" is our approach (with pixel constraint, $\ell_1$ loss and the changes to the model). The other columns are the standard models without any changes. EN = EfficientNet-B0, CN = ConvNeXt-tiny, RN = ResNet-18.

Table 2 shows that for the standard CNN the center of the image has the same effect as the corner. $\text{ERF}(1, 1)$ has the same value range as $\text{ERF}(\frac{h}{2}, \frac{w}{2})$. Compare this to our approach, where there is a large difference between the center and the corner ERF. More details are provided in the appendix in table A1 and table A2. A visual experiment in the appendix validates our approach with different positions (refer to fig. A1 and fig. A2).

### 4.2 Hypothesis: Top-GAP can be used as a size prior

Since our approach is concerned with limiting the number of pixels that a neural network can use, we evaluate sparsity using the $\ell_1$ matrix norm. The metric is defined as $\frac{1}{nm}||X||_1$ where $X = X^{(n+1)}$ is the normalized and upsampled feature map. We only select the ground truth channel. This is our CAM. Although this metric alone does not determine the quality of a CAM, it serves as an indicator of its noise level. In addition, a CAM with fewer highlighted pixels can facilitate the explanation of certain image features.

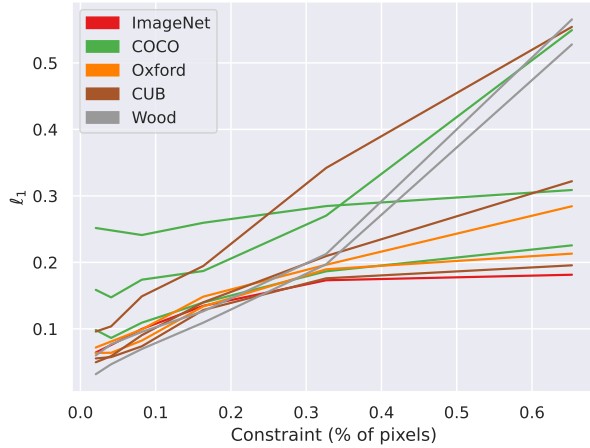

It is evident from fig. 3 that as we increase the constraint $k$, the number of displayed pixels on the CAM also rises (i.e. $||X||_1$ rises). This observation validates that our constraint effectively achieves the intended sparsity. While there are some fluctuations for certain datasets and architectures, the overall trend remains consistent.

Figure 3: Each line in the graph represents a dataset+architecture combination. The x-axis shows the normalized $k$ value (e.g. $\frac{64}{56^2}$) for the constraint, while the y-axis represents the $\ell_1$ norm. The constraint is given by the previously defined pooling layer Top-GAP$(\cdot, k)$.

Details of the results can be found in table A3. The table shows that we achieve higher sparseness than standard models. Furthermore, table A8 shows that simple $\ell_1$ regularization does not achieve the same level of sparsity.

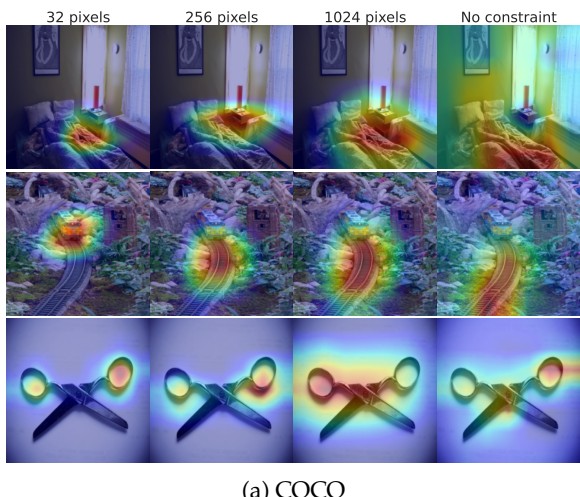 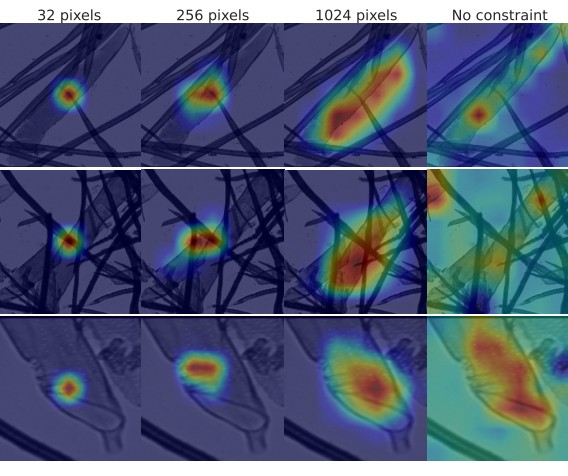

|                    |                      |
| :----------------: | :------------------: |
|     (a) COCO       | (b) Wood identification |

Figure 4: Impact of pixel constraint on CAM. The pixel constraint is always combined with $\ell_1$ loss and our new model structure. "No constraint" denotes a standard unmodified EfficientNet-B0 model using CAM/GradCAM (Selvaraju et al., 2019). For COCO: the ground truth for the first row is "bed". For the second row, it is "train" and for the third row, it is "scissors".

To visually demonstrate the effect of the constraint, we use the COCO dataset as an illustrative example. Consider the four images in the second row of fig. 4a showing a train on rails. In the first image, the constraint is 32 pixels and only the train is highlighted. In the 2nd through 4th images, either both the train and the rails or only the rails are highlighted. Since the ground truth class is "train", the rails should not be highlighted because the object ("train") itself best represents the class. With our approach (here: 32 pixels) it is possible to force the network not to use the feature "rails". This shows that our constraint allows to perform bias mitigation. The first and third row also show that different parts of the objects become more important. When not using any constraint, the CAM is affected by noise. This can be seen in the first row.

Figure 1 and fig. 4b show another example. The wood identification dataset with ConvNeXt-tiny was used for generating fig. 1. Given that the object size can be an important feature, the network attempts to capture this feature even with low constraint values. For instance, with the pixel constraint $k = 64$, we can observe that the network generated dots along the object's edges (see fig. 1a). This suggests that the network is attempting to figure out the object size by employing this strategy.

In the appendix, table A4 we show using a ground truth mask that our model focuses more on objects than background.

### 4.3   Hypothesis: higher robustness against simple PGD, FGSM and Square Attack

It was shown by Guo et al. (2018) that there exists a relationship between sparsity and adversarial robustness. Since we use the $\ell_1$ norm in our loss function, we expect increased robustness against certain adversarial attacks.

However, only applying $\ell_1$ to activations would greatly reduce clean accuracy. With our Top-GAP layer, we keep a high accuracy for datasets like ImageNet. Therefore, we do not set too many gradients to zero because this would make training the networks also harder ("shattered gradients" (Athalye et al., 2018)).

To assess this, we perturb the images in the datasets with FGSM and PGD using $\ell_\infty = 1/255$ (except for CIFAR-10 where we use $\ell_\infty = 8/255$). Then the predicted class of the perturbed images is compared with the class of the original images. The robust accuracy is the percentage of images where the prediction remains the same ("predicted class of perturbed image" equal to "predicted class of original image"). We use the library Foolbox (Rauber et al., 2017; 2020) for the adversarial attacks.

We tested for Top-GAP all $k$ and only report the value with the best clean accuracy and robust accuracy (FGSM, PGD). First, we consider datasets with higher resolution such as ImageNet.

| Dataset | Arch | FGSM ↑ | FGSM ↑ (ours) | PGD ↑ | PGD ↑ (ours) | Clean Acc | Clean Acc (ours) |
|---------|------|--------|---------------|-------|--------------|-----------|------------------|
| COCO | EN | 0.07 | **0.3063** | 0.0 | **0.1098** | $0.801 \pm 0.009$ | **0.803** $\pm 0.006$ |
| COCO | CN | 0.51 | **0.678** | 0.301 | **0.463** | $0.939 \pm 0.006$ | **0.940** $\pm 0.005$ |
| COCO | RN | 0.288 | **0.394** | 0.08 | **0.142** | $0.853 \pm 0.004$ | **0.868** $\pm 0.005$ |
| Wood | EN | 0.0 | **0.277** | 0.0 | **0.085** | $0.672 \pm 0.037$ | **0.681** $\pm 0.041$ |
| Wood | CN | 0.0 | **0.404** | 0.0 | **0.01** | $0.721 \pm 0.030$ | **0.724** $\pm 0.033$ |
| Oxford | EN | 0.037 | **0.107** | **0.0** | **0.0** | $0.854 \pm 0.008$ | **0.863** $\pm 0.010$ |
| Oxford | RN | 0.104 | **0.281** | 0.016 | **0.104** | $0.861 \pm 0.007$ | **0.862** $\pm 0.007$ |
| CUB | EN | 0.04 | **0.147** | 0.0 | **0.04** | $0.76 \pm 0.01$ | **0.77** $\pm 0.005$ |
| CUB | RN | 0.06 | **0.212** | 0.0 | **0.111** | **0.69** $\pm 0.014$ | $0.685 \pm 0.006$ |
| CUB | CN | 0.134 | **0.314** | 0.03 | **0.158** | **0.862** $\pm 0.007$ | $0.854 \pm 0.005$ |
| ImageNet | VG | 0.029 | **0.217** | 0.0 | **0.01** | **0.704** | 0.699 |
| ImageNet | RN | 0.065 | **0.256** | 0.0 | **0.059** | **0.698** | 0.697 |

Table 3: Our approach refers to the changed model with pixel constraint and $\ell_1$ loss. The original models come from PyTorch Image Models (Wightman, 2019) and are pretrained on ImageNet. EN = EfficientNet-B0, CN = ConvNeXt-tiny, RN = ResNet-18, VG = VGG11-bn.

For all the experiments in table 3, we use $\ell_\infty = 1/255$ (FGSM/PGD) and 40 steps (PGD). The $\pm$ sign denotes the standard deviation of the accuracy across 5 different folds. For ImageNet, we only report a single run due to computational complexity.

Next, we evaluate our method in table 4 on CIFAR-10.

| Method | Arch | PGD$^{20}$ ↑ | PGD$^{50}$ ↑ | Clean ↑ |
|--------|------|-------------|-------------|---------|
| Baseline | PRN18 | 0.0 | 0.0 | 0.945 |
| Top-GAP (ours) | PRN18 | 0.517 | 0.313 | **0.951** |
| FGSM-AT (Andriushchenko & Flammarion, 2020) | PRN18 | - | 0.476 | 0.81 |
| SAT (Peng et al., 2023) | RN50 | **0.552** | - | 0.849 |

Table 4: Results on CIFAR-10. We use $\ell_\infty = 8/255$ and 20/50 steps. SAT = Standard Adversarial Training, PRN18 = PreAct ResNet-18, RN50 = ResNet-50. Our results are close to the robustness of adversarially trained networks. Refer to appendix E for more experiments.

We see a much greater increase in adversarial robustness. Our method comes close to the results of adversarially trained networks for certain methods while maintaining high accuracy and high training speed. We repeated the experiments with the adversarial attacks of AutoAttack (Croce & Hein, 2020). While all attacks together lead to 0%, we achieve robustness against certain attacks. With PRN18, we achieve 34.26% with the square attack. For the adaptive APGD attacks, however, we see a decrease to about 7% (with 20 iterations). While this result is still much higher than the 0% achieved with a regular network, it is less than with an adversarial trained network.

Notably, square attack (Andriushchenko et al., 2020) does not rely on local gradient information. It should, therefore, be not affected by gradient masking. Nevertheless, we perform quite well against this attack; our model only suffers against adaptive attacks. This shows that our robustness is not necessarily a result of "shattered gradients" (Athalye et al., 2018).

Finally, we want to compare our method with other approaches that do not use adversarial training to achieve robustness. FLC Pooling (Grabinski et al., 2022) uses the 2D Fourier transform to achieve robustness, while k-WTA (Xiao et al., 2019) removes the top-k values after each layer.

| Method | Clean ↑ |
|--------|---------|
| k-WTA (Xiao et al., 2019) | $0.439 \pm 0.011$ |
| FLC Pooling (Grabinski et al., 2022) | $0.525 \pm 0.022$ |
| Top-GAP (ours) | **0.685** $\pm 0.006$ |

Table 5: Results on CUB with ResNet-18.

Table 5 shows that for real-world datasets like CUB, we achieve higher accuracy. Additionally, the use of the Fourier transform leads to slower training.

### 4.4 Hypothesis: higher robustness against distribution shifts

In addition to assessing adversarial robustness, it is valuable to analyze the network's performance under distribution shifts and potential biases. To address this, we use the Waterbirds dataset (Sagawa et al., 2019), where the backgrounds of images are replaced. Furthermore, we evaluate accuracy on ImageNet-Sketch (Wang et al., 2019) and ImageNet-C (Hendrycks & Dietterich, 2019). The results are in table 6.

| Dataset | Arch | Acc $\uparrow$ | Acc $\uparrow$ (ours) |
|---|---|---|---|
| CUB $\rightarrow$ Waterbirds | EN | 0.521 | **0.564** |
| CUB $\rightarrow$ Waterbirds | CN | 0.722 | **0.737** |
| CUB $\rightarrow$ Waterbirds | RN | 0.468 | **0.52** |
| ImageNet $\rightarrow$ Sketch | VG | 0.179 | **0.20** |
| ImageNet $\rightarrow$ Sketch | RN | 0.206 | **0.236** |
| ImageNet $\rightarrow$ ImageNet-C | VG | 0.494 | **0.498** |
| ImageNet $\rightarrow$ ImageNet-C | RN | 0.513 | **0.535** |

Table 6: Evaluation of the out-of-distribution accuracy by using images outside the original dataset. $X \rightarrow Y$ means train on X and validate on Y. For instance, we trained an EfficientNet-B0 (EN) model on the CUB dataset and then assessed its accuracy on the Waterbirds dataset. For ImageNet-C (Hendrycks & Dietterich, 2019), we use strength level 1 and take the average of all types of corruption.

A slight improvement in accuracy can be observed for all datasets. While there are many works that show higher accuracy for datasets such as ImageNet-Sketch Fang et al. (2022), they are based on specialized training methods (self-supervised, semi-supervised) and/or more data. Our proposed method comes "without cost" in the sense that it works for any architecture and dataset, without requiring more GPU resources. It can be viewed as a regularization technique as well.

### 4.5 Ablation studies

Having established the effectiveness of our approach across various datasets and architectures, our next objective is to assess the impact of the individual components within our solution. We aim to determine whether each component is essential or if certain components can be omitted while still maintaining satisfactory performance. Furthermore, we want to show that the increase of accuracy as seen in table 3 is not a consequence of having a slightly higher number of parameters.

We perform an ablation study on the COCO dataset. There are in total $2^3 = 8$ possibilities as can be seen in table 7.

| FPN | $\ell_1$ loss | Top-GAP | $Acc_{COCO}$ $\uparrow$ | $FGSM_{COCO}$ |
|---|---|---|---|---|
| ✗ | ✗ | ✗ | 0.801 | 0.07 |
| ✗ | ✗ | ✓ | 0.681 | 0.0 |
| ✗ | ✓ | ✗ | 0.799 | 0.297 |
| ✗ | ✓ | ✓ | 0.796 | 0.263 |
| ✓ | ✗ | ✗ | 0.796 | 0.082 |
| ✓ | ✗ | ✓ | 0.532 | 0.054 |
| ✓ | ✓ | ✗ | 0.790 | 0.305 |
| ✓ | ✓ | ✓ | **0.803** | **0.306** |

Table 7: Ablation study using the COCO dataset and EfficientNet-B0. The $\ell_1$ regularization is beneficial for robustness, but only the combination of all three components also leads to improvements in localization (while maintaining accuracy). We repeated the experiments with the Wood dataset and came to the same results.

When FPN is deactivated, we set the pixel constraint to $k = 4$ for Top-GAP since the final feature matrix has dimensions $7^2$. However, when FPN is activated, we increase the constraint to $k = 256$ because the final feature matrix is larger due to upsampling (size $56^2$). The ratio is the same e.g. $\frac{1024}{56^2} = \frac{16}{7^2}$.

Table 7 shows that adding a FPN module to the architecture does not consistently increase the accuracy. Only a combination of multiple components leads to an increase. Furthermore, $\ell_1$ regularization alone would not lead to better interpretability/robustness (refer to appendix table A7).

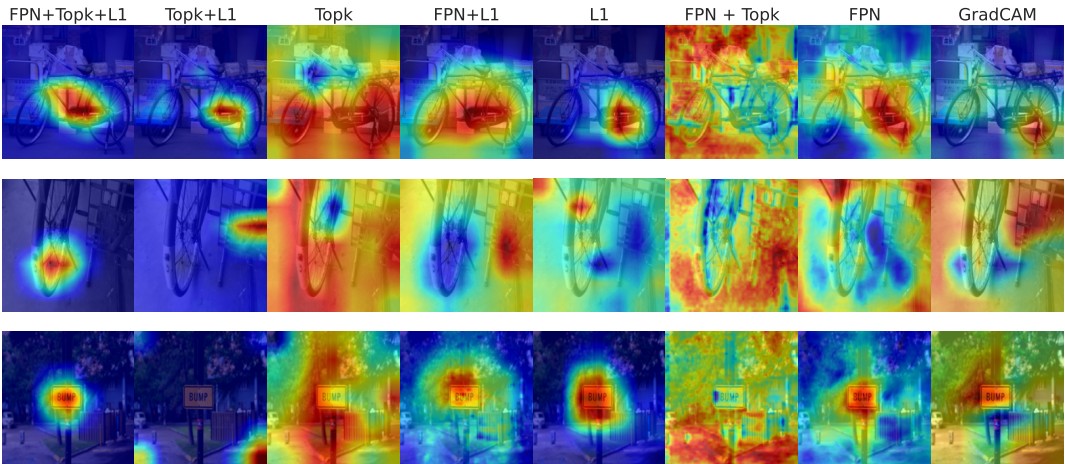

Figure 5: Visual ablation study on COCO. The first two rows show a bicycle, while the third one shows a sign. Only the variant FPN + Top-GAP + $\ell_1$ localizes all three objects correctly. The models that were trained with some kind of $\ell_1$ regularization tend to have less noise.

Although our approach has shown improved accuracy, it is still critical to visually assess the impact of the constraints. To illustrate this, refer to fig. 5. Here we compare all eight configurations using three images from the COCO dataset. The figures clearly show that the combination of the three components: FPN, $\ell_1$-loss, and Top-GAP, provides the most favorable results. The figure also shows that the inclusion of some form of $\ell_1$-loss generally improves the quality of the CAMs by reducing noise.

## 5  Discussion and Outlook

In this paper, we presented a new approach to improve the native robustness of CNNs. Depending on the dataset and architecture, we see major improvements against simple PGD, FGSM and Square Attack. Our method has no negative effects like other techniques (accuracy, speed, memory). It is well suited to defend against fast attacks. Slower ensemble attacks such as AutoAttack are computationally more expensive.

Our method focuses on controlling the number of pixels a network can use for predictions, resulting in CAMs with lower noise and better localization. The results show that our approach is effective on a variety of datasets and architectures. We have consistently observed both visually and numerically more concise feature representations in the CAMs. In addition, our approach provides a novel form of network regularization. By forcing the network to focus exclusively on objects of a predefined size, we reduce the risk of highlighting irrelevant regions, which can be critical for applications that require precise object localization or for reducing bias.

**Limitations.** Determining the optimal value for the pixel constraint parameter $k$ currently depends on hyperparameter tuning. It is possible to explore automated methods for determining this parameter to improve efficiency and adaptability. Second, given the variety of object sizes, it may not be ideal to rely on a single parameter for all objects. Only in specific areas such as biomedical imaging, where object size are not influenced by perspective projections (e.g. microscope) typically show low size variances. Investigating ways to dynamically adjust this parameter for different object sizes would be a valuable line of research. Finally, the proposed FPN module can be further refined to improve accuracy even more.

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

# Top-GAP: Integrating Size Priors in CNNs for more Robustness, Interpretability, and Bias Mitigation

## Supplementary Material

The appendix contains the following additional materials:

- A detailed description of the datasets.

- More details regarding the effective receptive field: table A1, table A2. Qualitative analysis of the ERF with fig. A1 and fig. A2.

- More details regarding sparsity: table A3.

- An experiment verifying the overlap with human annotations: table A4.

- More experiments for the CIFAR datasets: appendix E

- Effect of $\ell_1$ normalization on robustness and interpretability: table A7, table A8 and table A9.

## A    Description of datasets

We test all our models on the following datasets:

- COCO (Lin et al., 2015): We turned this segmentation dataset into a classification dataset by excluding images with more than one object. Furthermore, we kept only classes with a minimum of 20 samples per class. The resulting subset comprises 53 classes.

- Wood identification dataset (Nieradzik et al., 2023): This dataset consists of high-resolution microscopy images for hardwood fiber material. Nine distinct wood species have to be distinguished.

- Oxford-IIIT Pet Dataset (Parkhi et al., 2012): The task is to differentiate among 37 breeds of dogs and cats.

- CUB-200-2011 (Wah et al., 2011) and Waterbirds (Sagawa et al., 2019): 200 classes of birds have to be distinguished. Waterbirds replaces the background of the original images to test the models for biases.

- ImageNet (Deng et al., 2009): A large-scale dataset with 1000 different classes. ImageNet-Sketch (Wang et al., 2019) / ImageNet-C (Hendrycks & Dietterich, 2019) replaces the original validation images with out-of-distribution / corrupted images.

- CIFAR10: A dataset where each image has a size of $32 \times 32$. 10 classes have to be distinguished.

# B  Effective Receptive Field (ERF)

## B.1  Quantitative analysis

In table 2, we only showed the difference between the center and the corner ERF. In the following tables, we provide the individual values. The gradients were z-normalized to have mean at $0$ and standard deviation at $1$.

| Dataset | Arch | Center ERF $\uparrow$ | Center ERF (ours) $\uparrow$ |
|---|---|---|---|
| COCO (Lin et al., 2015) | EN | 0.534 | 0.54 |
| COCO (Lin et al., 2015) | CN | 0.47 | 0.439 |
| COCO (Lin et al., 2015) | RN | 0.595 | 0.571 |
| Oxford (Parkhi et al., 2012) | EN | 0.087 | 0.51 |
| Oxford (Parkhi et al., 2012) | RN | 0.104 | 0.542 |
| CUB-200-2011 (Wah et al., 2011) | EN | 0.489 | 0.493 |
| CUB-200-2011 (Wah et al., 2011) | CN | 0.477 | 0.398 |
| CUB-200-2011 (Wah et al., 2011) | RN | 0.538 | 0.534 |
| Average | - | 0.412 | **0.503** |

Table A1: Center ERF. "Ours" is our approach (with pixel constraint, $\ell_1$ loss and the changes to the model). The other columns are the standard models without any changes. EN = EfficientNet-B0, CN = ConvNeXt-tiny, RN = ResNet-18.

The table shows that for the center pixel the gradient with respect to the input image is higher, when using our method.

| Dataset | Arch | Corner ERF $\downarrow$ | Corner ERF (ours) $\downarrow$ |
|---|---|---|---|
| COCO (Lin et al., 2015) | EN | 0.426 | 0.093 |
| COCO (Lin et al., 2015) | CN | 0.398 | 0.151 |
| COCO (Lin et al., 2015) | RN | 0.322 | 0.172 |
| Oxford (Parkhi et al., 2012) | EN | 0.074 | 0.127 |
| Oxford (Parkhi et al., 2012) | RN | 0.044 | 0.099 |
| CUB-200-2011 (Wah et al., 2011) | EN | 0.522 | 0.013 |
| CUB-200-2011 (Wah et al., 2011) | CN | 0.511 | 0.156 |
| CUB-200-2011 (Wah et al., 2011) | RN | 0.446 | 0.005 |
| Average | - | 0.343 | **0.102** |

Table A2: Corner ERF. The values are lower using our approach.

Similarly, we see that the pixels have less of an effect when the corner of the input image is considered.

## B.2   Qualitative analysis

After having analyzed numerically the receptive field for positions corner $= (1, 1)$ and center $= (4, 4)$, we also want to visually analyze the effect on the ERF.

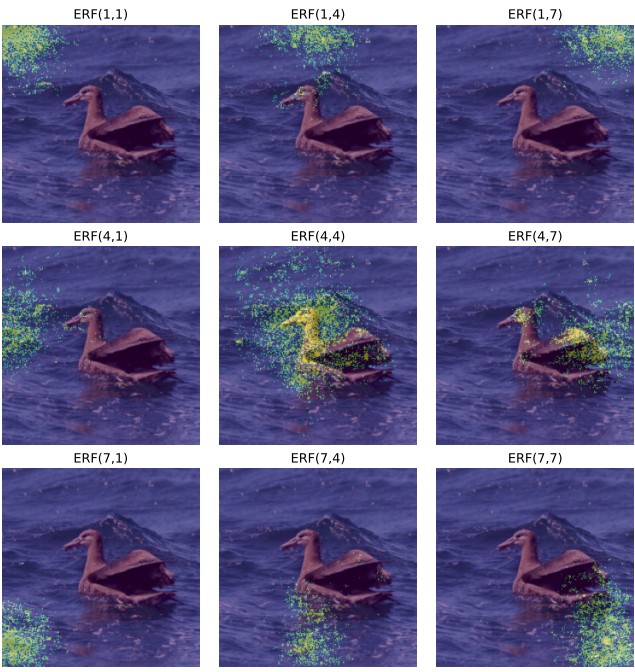

Figure A1: Standard ResNet-18 (dataset: CUB)

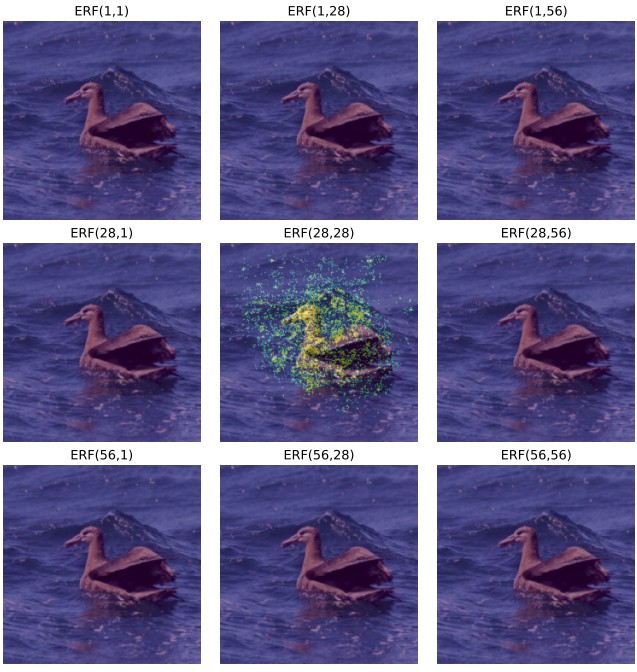

Figure A2: ResNet-18 with our approach (dataset: CUB)

Comparing fig. A1 and fig. A2, we see that the background has less of an effect using our approach.

## C Sparsity

For almost all datasets and architectures, our approach achieved sparser CAMs. We see especially large decreases for ImageNet.

| Dataset | Arch | $\ell_1 \downarrow$ | $\ell_1 \downarrow$ (ours) |
|---|---|---|---|
| COCO (Lin et al., 2015) | EN | 0.179 | **0.064** |
| COCO (Lin et al., 2015) | CN | 0.251 | **0.151** |
| COCO (Lin et al., 2015) | RN | **0.173** | 0.194 |
| Wood (Nieradzik et al., 2023) | EN | 0.190 | **0.032** |
| Wood (Nieradzik et al., 2023) | CN | 0.110 | **0.046** |
| Oxford (Parkhi et al., 2012) | EN | 0.154 | **0.072** |
| Oxford (Parkhi et al., 2012) | RN | 0.151 | **0.064** |
| CUB-200-2011 (Wah et al., 2011) | EN | 0.235 | **0.05** |
| CUB-200-2011 (Wah et al., 2011) | CN | 0.164 | **0.096** |
| CUB-200-2011 (Wah et al., 2011) | RN | 0.121 | **0.056** |
| ImageNet (Deng et al., 2009) | VG | 0.279 | **0.064** |
| ImageNet (Deng et al., 2009) | RN | 0.387 | **0.123** |

Table A3: The last column reports the sparsity of the CAM using our approach (with pixel constraint, $\ell_1$ loss and the changes to the model). The third column is a standard model without any changes. For the standard model, we use GradCAM. EN = EfficientNet-B0, CN = ConvNeXt-tiny, RN = ResNet-18, VG = VGG11-BN.

Only the lowest $\ell_1$ of the different $k$ values is reported. We observe that in general a strong pixel constraint such as $k = 64$ pixels leads to the lowest $\ell_1$ value.

## D Human annotation

Although neural networks may prioritize different regions compared to humans, segmentation masks remain valuable sources of information. For example, if the network focuses on the background instead of the relevant object, it suggests potential classification errors when the object appears no longer with the same background.

The segmentation masks serve as the "ground truth" in our analysis of the COCO and CUB datasets. We compute the pixel-wise intersection over union (IOU) to identify the predicted mask that has the largest overlap with the ground truth. Since standard CNN models do not inherently generate a class activation map, we use the GradCAM method to generate the predicted mask in this context. The results are in table A4.

| Dataset | Arch | IOU $\uparrow$ | IOU $\uparrow$ (ours) |
|---|---|---|---|
| COCO | EN | 0.309 | **0.348** |
| COCO | CN | 0.103 | **0.361** |
| COCO | RN | 0.371 | **0.391** |
| CUB | EN | 0.323 | **0.414** |
| CUB | CN | 0.125 | **0.389** |
| CUB | RN | 0.268 | **0.435** |

Table A4: The last column is our approach ("ours"). The third column is a standard unchanged model. For the standard model, we use GradCAM.

In all cases, our approach consistently demonstrates a higher IOU. Interestingly, ConvNeXt-tiny exhibits a more pronounced improvement ($\approx 25\%$) with our approach compared to the other architectures. The best $k$ value depends here on the actual size of the object. Since the objects of the CUB and COCO datasets are relatively large, we need higher $k$ values.

# E More CIFAR-10 experiments

## E.1 Effect of steps on PGD

We analyze the effect of the parameter `steps` on the function `LinfProjectedGradientDescentAttack` of Foolbox. Three models are compared: Andriushchenko2020Understanding (Andriushchenko & Flammarion, 2020), baseline (standard PreAct ResNet-18), Top-GAP (our approach). We sampled 500 images from the dataset.

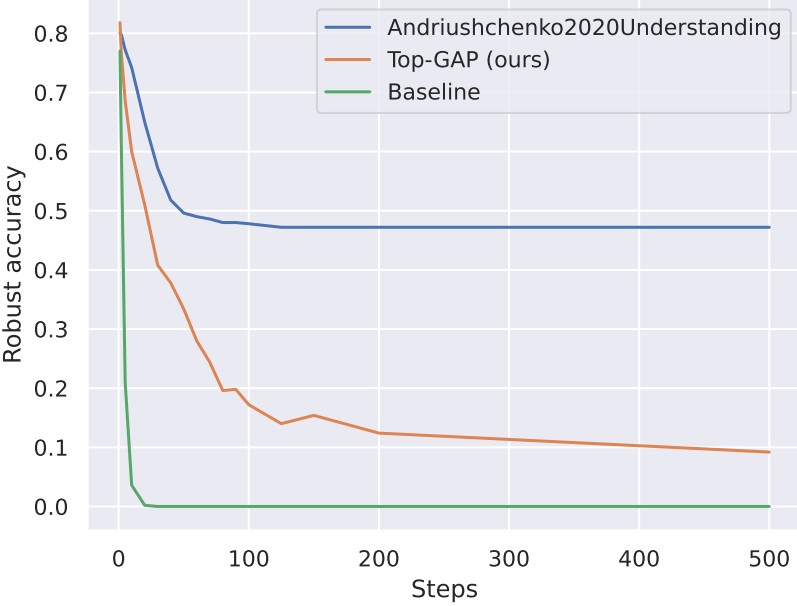

Figure A3: Effect of steps on $\ell_\infty$-PGD

## E.2 Architectures

The experiments from the main paper were repeated for different architectures. Furthermore, the hyperparameters were optimized for robust accuracy. The results were averaged across 5 different seeds.

| Arch | PGD$^{20}$ ↑ | PGD$^{50}$ ↑ | Clean Acc ↑ |
|------|------|------|------|
| PreAct-ResNet18 | **0.5484** $\pm$ 0.0135 | **0.3528** $\pm$ 0.0277 | 0.9493 $\pm$ 0.0013 |
| ResNet18 | 0.5407 $\pm$ 0.033 | 0.3361 $\pm$ 0.0516 | 0.9501 $\pm$ 0.0009 |
| ResNet34 | 0.4254 $\pm$ 0.08 | 0.2989 $\pm$ 0.1181 | 0.9513 $\pm$ 0.0023 |
| ResNet50 | 0.4815 $\pm$ 0.0299 | 0.336 $\pm$ 0.0462 | **0.9515** $\pm$ 0.0023 |
| WideResNet40-4 | 0.4156 $\pm$ 0.0092 | 0.2811 $\pm$ 0.0253 | 0.9462 $\pm$ 0.0002 |

Table A5: We use $\epsilon = {}^8/_{255}$ and 20/50 steps.

## E.3 Combination with Adversarial Training

Next, we test whether our approach can be combined with adversarial training. Since adversarial training requires extensive computational resources, only CIFAR-10 is tested.

For our evaluation, we use the RaWideResNet-70-16 model (Peng et al., 2023), which represents the current state-of-the-art on RobustBench. This model was trained with an additional 50 million synthetic images under the $(\ell_\infty, \epsilon = \frac{8}{255})$ threat model. Then we modified this architecture by adding our FPN module, the $\ell_1$ loss and the Top-GAP pooling layer. Only the layers of the FPN module were trained, while all other layers

were frozen. No adversarial training was used for finetuning. Finally, we compare this modified model with the standard model.

| Arch | PGD$^{20}$ ↑ | PGD$^{50}$ ↑ | Clean Acc ↑ |
|------|-----------|-----------|-------------|
| RaWideResNet-70-16 | **0.8494** | **0.7462** | 0.9372 |
| RaWideResNet-70-16 + ours | 0.8463 | 0.7168 | **0.953** |

Table A6: The numbers of the standard model differ slightly from the numbers in the original paper because we evaluated all 50,000 images of the test set. We again use $\epsilon = {}^8/_{255}$.

While we see a slight decrease in robustness of around 3%, the accuracy increases when using our approach. There is an improvement of around 2%.

## F    Effect of $\ell_1$ normalization on robustness and interpretability

From the ablation study in table 7, we have seen that the $\ell_1$ regularization has a strong influence on the results. Here we want to show that without the other components we would have lower interpretability, robustness and/or accuracy.

We consider the following variants of ResNet-18:

- $\ell_1$ regularization only on the last feature output (activations) with $\lambda = 1.0$

- $\ell_1$ regularization only on the last feature output (activations) with $\lambda = 0.1$

- $\ell_1$ regularization on all activations with $\lambda = 10^{-3}$

- $\ell_1$ regularization on all activations with $\lambda = 10^{-5}$

- ours: our approach

We use the CUB-200-2011 dataset. $\lambda$ denotes the strength of the regularization.

### F.1   Interpretability and accuracy

First, we analyze the effective receptive field and accuracy.

| Approach | $\lambda$ | Center ERF ↑ | Corner ERF ↓ | Accuracy |
|----------|-----------|--------------|--------------|----------|
| last | 1.0 | 0.514 | **0.002** | 0.63 |
| last | 0.1 | **0.536** | 0.113 | **0.69** |
| all | $10^{-5}$ | 0.488 | 0.416 | **0.69** |
| all | $10^{-3}$ | 0.335 | 0.26 | 0.19 |
| ours | 1.0 | **0.534** | **0.005** | **0.69** |

Table A7: $\ell_1$ regularization is a tradeoff between accuracy and ERF for the other approaches.

We see that we are only able to influence the ERF by regularizing the last feature map. While the approach "last + $\lambda = 1.0$" also achieves the same ERF as "ours", we see a significant decrease in accuracy of about 6%. Instead, we can also decrease $\lambda$, then the accuracy is the same, but we lose interpretability.

Additionally, without our Top-GAP pooling, we can no longer control the number of pixels. The $\lambda$ parameter cannot be used for that.

Let $X$ be the last feature output. We measure how many pixels are highlighted in the output, when adjusting $\lambda$ and our Top-GAP $k$ pixel constraint.

| Approach | $\lambda$ | $||X||_1$ | Accuracy ↑ |
|---|---|---|---|
| last | 1.0 | 0.141 | 0.63 |
| last | 0.1 | 0.152 | **0.69** |
| all | $10^{-5}$ | 0.119 | **0.69** |
| all | $10^{-3}$ | 0.389 | 0.19 |
| ours, $k = 128$ | 1.0 | 0.057 | 0.66 |
| ours, $k = 256$ | 1.0 | 0.072 | 0.67 |
| ours, $k = 512$ | 1.0 | 0.126 | 0.68 |
| ours, $k = 1024$ | 1.0 | 0.174 | **0.69** |
| ours, $k = 2048$ | 1.0 | 0.193 | 0.68 |

Table A8: As we increase the constraint value $k$, the number of pixels increases. The same behavior is not possible using $\lambda$. The accuracy would suffer too much.

When we increase the regularization strength from $\lambda = 0.1$ to $\lambda = 1.0$, the number of pixels only decreases from $0.152$ to $0.141$. However, the accuracy decreases by $6\%$.

Compare this to our approach. We can decrease the number of pixels while keeping the accuracy at the same level.

## F.2 Robustness

Next, we analyze the level of robustness with respect to $\ell_1$ regularization.

| Approach | $\lambda$ | PGD$^{40}$ ↑ | FGSM |
|---|---|---|---|
| last | 1.0 | 0.06 | 0.15 |
| last | 0.1 | 0.03 | 0.11 |
| all | $10^{-5}$ | 0.0 | 0.06 |
| all | $10^{-3}$ | 0.0 | 0.03 |
| ours | 1.0 | **0.11** | **0.21** |

Table A9: Regularizing the last layer leads to the highest level of robustness. Our approach surpasses a simple regularization.

Regularizing only the last layer also brings a certain degree of robustness, but it comes at a price. The accuracy is lower and we still do not achieve the same level of sparsity for $\lambda = 1.0$ as with our approach.

