# OpenReview forum: "Top-GAP: Integrating Size Priors in CNNs for more Robustness, Interpretability, and Bias Mitigation"
_TMLR — Rejected by TMLR_

### Review · Reviewer_5Kpo · 2024-05-02

**Summary Of Contributions:**

The submission presents an architectural modification, named Top-GAP, aimed at improving the robustness and explainability of convolutional networks for vision applications.
The proposal features three components: feature pyramid networks, l1 regularization on the last feature map, and a top-k pooling layer.
The authors present results on robustness and interpretability, in terms of IoU for segmentation labels over the class activation maps (CAM).

**Audience:**

Yes

**Broader Impact Concerns:**

No concerns.

**Claims And Evidence:**

No

**Requested Changes:**

- Add an evaluation using AutoAttack [3] to measure adversarial robustness, and more generally report results against stronger attacks (and epsilons larger than 1/255) in the main body of the paper;
- Provide GradCAM-based results in Table 7 for the proposed approach, and compare against other architecture-based CAM schemes;
- Generally tone down the claims on improved robustness and interpretability.

### References

[3] Reliable evaluation of adversarial robustness with an ensemble of diverse parameter-free attacks, Croce and Hein, ICML 2020

**Strengths And Weaknesses:**

### Strengths

The paper tackles a rather interesting subject: architectural modifications towards robustness and interpretability.

### Weaknesses

The paper lacks important details and definitions, that are important to fully assess and understand the contributions and proposal. Notably, the paper lacks a background section, which should feature a detailed description of both CAM and GradCAM. For instance, concerning CAM, reference [1] appears to be pretty relevant: how is the proposed "CAM mode" of the model related to [1]?

Even more importantly, I am not convinced that the proposed experimental results provide enough evidence concerning the fact that the proposed approach increases model robustness or interpretability.

Concerning interpretability, given the use of $\ell_1$ regularization on the $X$ activations, I do not think that the sparsity in Table 2 is any surprising. Table 7, instead, is comparing an architecture featuring architectural CAM (another example is [1]) with architecture-agnostic GradCAM. What if GradCAM was used for the proposed approach?

Concerning robustness, the increased resistance to adversarial attacks appears to be mostly linked to the attacks' weakness, rather than  to true empirical robustness. This is clearly evidenced by Figure A2 in appendix B, where --differently from the adversarially trained network-- robustness starkly decreases with the number of PGD steps. The remaining robustness could very well be simply gradient obfuscation. Indeed, when combined with adversarial training, the proposed architecture results in a robustness decrease.
Furthermore, as highlighted by the ablation study, robustness appears to be mostly linked to the use of $\ell_1$ regularization on the $X$ activations. Similar results correlating activation sparsity and robustness are for instance presented in [2].
Finally, other heuristic defences (not based on adversarial training) against adversarial robustness are not compared against.


### References

[1] Learning Deep Features for Discriminative Localization, Zhou et al., CVPR 2016

[2] Sparse DNNs with Improved Adversarial Robustness, Guo et al., NeurIPS 2018

---

> ### Author Response · Authors · 2024-05-24
>
> Thank you for feedback. We agree that our experiments needed more clarity. We added more experiments and also did substantial other changes to the document. Refer to the general comment above for the list of all changes.
>
> **Lack of terminology**
>
> We rewrote the introduction of the Methods section to explain in more detail certain terms.
>
> The paper "Learning Deep Features for Discriminative Localization" introduces the standard approach for CAMs. This is in our new section definition 3.3.
>
> The "CAM mode" of the standard approach is equivalent to our "CAM mode". The difference is that during training we modify the model weights (L1 regularization, architecture changes and pooling).
>
> Approaches such as "CAM", GradCAM or LayerCAM differ how they compute the weights during prediction. There are no changes during training. Therefore, they do not affect accuracy, robustness etc. These approaches cannot change the interpretability on a more fundamental level. The results are fixed by the model weights.
>
> **Interpretability**
>
> We have moved the previous table 2 to the appendix. We added a new section 4.1 to show the effect of pixels on the classification.
>
> While we only regularize the activations in the last layer, this sparseness affects all the layers. This leads to changes in the interpretability.
>
> By introducing the section 4.1, we show that the effective receptive field focuses more on the center, where the object is also located.
>
> Due to time and page constraints, we decided to move the previous table 7 to the appendix. However, we note that if we were to use LayerCAM, GradCAM or any other of the approaches in combination with our model, it would barely show any differences. CAMs tend to perform quite similarly, refer to example [1]. Most papers in the CAM literature focus on old architectures and datasets that do not reflect the real world. With our approach, we can show more substantial differences.
>
> **Robustness**
>
> We have added several new experiments to show the increase in robustness. See reviewer uKRt “Gradient Masking”.
>
> The square attack does not rely on gradient information. We achieve more than 34% robustness on CIFAR-10. Only against adaptive attacks we have lower robustness.
>
> While the increase in robustness is mainly related to the l1 regularization, the other components also lead to an increase in robustness. More importantly, we want to maintain the same accuracy. When using only l1 sparsity, we found large accuracy degradation for ImageNet.
>
> Other non-AT approaches tend to perform worse as they destroy the gradients or make training the network more difficult. This can be seen in Table 5, where the accuracy drops by more than 30%. The reason for this is that most approaches in the literature are only evaluated on ImageNet and CIFAR-10.
>
> In contrast, our approach also works for real-world datasets without reducing the training speed.
>
> [1] https://github.com/frgfm/torch-cam/releases/download/v0.3.1/example.png

---

> > ### Comment · Reviewer_5Kpo · 2024-05-30
> >
> > Thank you for your response. I appreciate the clarifications concerning background and terminology.
> > However, I am still not convinced about the claims made on both interpretability and robustness.
> > - Could the authors please empirically compare against the use of  $\ell_1$ regularization on standard networks, on both the weights and the activations (and not only the last activations, but throughout the network), for what concerns both robustness and interpretability? The current ablation already suggests that the proposed $\ell_1$ loss (a common activation sparsity regularizer, on features) does most of the work concerning robustness and interpretability.
> > - Concerning AutoAttack accuracy, the authors should report, as commonly done in the literature, the accuracy after going through the standard evaluation as listed in the original paper. The fact that APGD already voids most of the robustness of the network is concerning, in this regard. I see that the authors have now rephrased to "*robustness against non-adaptive attacks*", but I am not exactly sure this is a scientifically sound statement. A carefully-tuned PGD attack, with enough restarts and steps, will likely get to the same performance of what the authors refer to adaptive attacks. The current focus on weak attacks will give a false sense of security, mainly to readers outside of the research area. I would suggest that the authors provide further clarifications in the text regarding this.

---

> > > ### Author Response · Authors · 2024-06-03
> > >
> > > Thank you for taking the time to read our revision. We have added additional experiments to the appendix: section F (last two pages). The changed text was highlighted in blue. We are happy to make any further changes or conduct additional experiments as needed.
> > >
> > > ## Interpretability
> > >
> > > We show that only the $\ell_1$ regularization on the last activations is useful. However, the standard l1 regularization is still not able to control sparsity to the same extent. Increasing lambda in “loss += lambda * l1” reduces the number of used pixels only slightly, but at the same time the accuracy deteriorates many times over. We are unable to get the same level of sparsity in the CAM.
> > >
> > > Therefore, the lambda term is not equivalent to our Top-GAP "k pixel constraint" term.
> > >
> > > Applying l1 regularization to all terms, even at values such as $10^{-3}$ or $10^{-5}$, does not change the effective receptive field.
> > >
> > > ## Robustness
> > >
> > > We repeat the experiment for PGD and FGSM. Here, too, we see that regularization of the last layer leads to the best results. However, the results are still worse than our approach.
> > >
> > > The problem is that the value of lambda needs to be high to be useful, but then the clean accuracy deteriorates too much. Increasing lambda from 0.1 to 1.0 in the standard model, reduces $||X||_1$ by 0.01 while accuracy decreases by 6% ($X$ is the CAM). In comparison, decreasing our k from 2048 to 128, reduces $||X||_1$ by 0.14 and decreases accuracy only by 3%.
> > >
> > > ## AutoAttack accuracy
> > >
> > > We have changed the term “non-adaptive” to “simple PGD, FGSM, Square Attack”. This should make it clear to the reader that we do not claim to be state-of-the-art, but rather offer robustness against these specific attacks. We have also added a sentence emphasizing that all attacks of AutoAttack together would result in 0% robustness. So our contribution with this paper is to provide robustness against certain attacks, keep the same (or higher) accuracy and have better interpretability (enables the user to control the number of pixels).
> > >
> > > We have also added another experiment that visually explains the effect on the ERF (see appendix B.2). This confirms again our results.

---

### Review · Reviewer_dPK1 · 2024-05-02

**Summary Of Contributions:**

The authors aim to limit the number of pixels that are relevant for the class-prediction by limiting the number of features of the last convolutional layer to $k$ features. The limitation is performed via an altered global average pooling (GAP) layer, called Top-GAP, and an addition of a $L_1$ regularization term of the feature map to the cross-entropy loss. The proposed method is mainly evaluated in terms of adversarial robustness to FGSM and PGD attacks, and sparsity of the Class Activation Maps (CAMs).

**Audience:**

Yes

**Claims And Evidence:**

No

**Requested Changes:**

* Please introduce your terminology. There is no explanation of a CAM, and no definition of terms like "size of feature representations", "locations in feature map", "empty feature map", "specific-sized features", the size of a CAM, etc.
* Discuss the connection between the limited feature map expressions via the Top-GAP layer and the number of used pixels from the input image. Under which assumptions and to what extend does the Top-GAP layer theoretically reduce the number of relevant pixels? I don't follow the calculations about the pixel constraint after Eq. (2) at all.
* Focus the experiments on specific hypotheses (e.g., the proposed method increases adversarial robustness in such and such a way). What is the purpose of the experiment from Table 5? I assume it's about generalizability, but then you should compare to other related methods. Would it not be reasonable to assume that other related methods also increase the adversarial robustness?

**Strengths And Weaknesses:**

## Strengths
* Good motivation to limit the amount  of pixels used for classification
* It's interesting to evaluate a classifier using fewer pixels with regard to adversarial robustness
* Good literature overview
## Weaknesses
* The method relies on the CAM assumption that we can map importance from feature maps back to the input pixels, which is unclear.
* The empirical evaluation tries to do too many things at once, the provided experiments are very shallow and not very conclusive
* Missing explanations and definitions of the used terminology make it hard to understand what the authors really mean, reproducibility given the text is low.

The promise of the paper, to reduce the amount of pixels being used for classification, makes a lot of sense to me. It seems like a natural step to regularize neural networks towards a recognition of images that is closer to human perception. In particular, if we can control the amount of relevant pixels, then we might also be able to identify those relevant pixels, which would be indeed a big step towards trustworthiness and explainability. However, I don't really see this promise being fulfilled, when the method relies on XAI methods that in turn rely on unproven assumptions. I don't see that we can guarantee that a classifier with the proposed restrictions on the feature map does indeed reduce the number of pixels used for classification. The provided evaluation relies on visual inspection of XAI methods like CAM and Grad-CAM that are to be taken with a grain of salt [1]. A more convincing analysis would have been to observe whether the adversarial attacks indeed focus on the areas that are deemed as relevant by the method, or if it's, for example, possible to change the class prediction by attacking only "irrelevant" pixels. The provided empirical analysis is rather shallow in the sense that it provides limited insight. The experiments show sometimes a notable increase, sometimes no increase in adversarial robustness. Especially because the authors compare the gain in robustness to adversarial training, I would have expected a stronger result. A comparison to adv. training is only given for CIFAR-10 and the PGD attack (using infinity norm?). The results are fine, but many details are unclear (why is there only one PGD evaluation for the AT trained models? Which adversarial attack is used for SAT?). Generally, the authors don't compare their method to any of the other methods that also try to improve the focus of the classifier to specific/smaller regions, hence, there is no empirical indication of strengths and weaknesses of the proposed method.

[1] Viering, Tom, et al. "How to manipulate cnns to make them lie: the gradcam case." arXiv preprint arXiv:1907.10901 (2019).

---

> ### Author Response · Authors · 2024-05-24
>
> Thank you for feedback. We have put more emphasis on robustness, restructured the paper and added some of the requested experiments. Refer to the general comment above for the list of all changes.
>
> **Mapping importance from feature map back to input**
>
> We added a new experiment using the effective receptive field (ERF) to show that fewer input pixels affect the  corner of the image.
>
> The theoretical receptive field [2] covers the entire image. In practice, however, the effective receptive field is much lower. With our method, it drops even further.
>
> **Too many things at once**
>
> We moved several experiments to the appendix to focus more on robustness and the ERF.
>
> **Unclear terminology**
>
> We rewrote the introduction of the Method section. Previously, we did not define exactly how the pixels in the feature map are related to the input. Furthermore, we now give a more precise definition of class activation map and GradCAM.
>
> **Why is there only one PGD evaluation for the AT trained models?**
>
> We did not train ourselves the FGSM-AT and SAT model (Table 4). We use the numbers provided by the two authors. The reason is that even for CIFAR-10 adversarial training requires a long time.
>
> **Not always increase in adversarial robustness**
>
> Whether we see an increase in robustness depends on two factors: (1) strength of the attack, (2) method type. While we do not get increased robustness for every method, our robustness comes without a cost (refer to reviewer uKRt "Gradient masking"). We have slightly reduced our claims to focus on non-adaptive robustness.
>
> Furthermore, certain adversarial attacks are so strong that even with the human eye, it would be easy to detect them. Achieving robustness against these, is not a necessity.
>
> **Comparison with other methods**
>
> Many methods that claim to improve the focus of the classifier show it based on accuracy. For example, [3] introduced the Squeeze-and-Excitation layer, which is notably used in EfficientNet. It should act as an "attention" mechanism. When looking at the datasets trained on EfficientNet, we do not see a difference with regard to the ERF (Table 2, A1, A2).
>
> **Focus on hypotheses**
>
> We restructured the experiments. Furthermore, we added a comparison with FLC Pooling and k-WTA (Table 5).
>
> [1] Understanding the effective receptive field in deep convolutional neural networks https://arxiv.org/abs/1701.04128
> [2] Computing Receptive Fields of Convolutional Neural Networks https://distill.pub/2019/computing-receptive-fields/
> [3] Squeeze-and-Excitation Networks https://arxiv.org/pdf/1709.01507

---

> > ### Comment · Reviewer_dPK1 · 2024-06-01
> > **Review of Revision**
> >
> > Dear authors,
> > Thanks for putting effort into making the paper more focused and clear. Unfortunately, I still have many open questions about the newly added text (see below). On a higher level, I don't see the main weaknesses resolved. The paper still lacks focus on its contribution and I would still say that reproducibility is low. The experiments don't sufficiently show that there is empirically a robustness increase to be expected from the proposed approach, I agree with the assessment of Rev. 5Kpo here. Regarding the influence on the importance of pixels, I consider the new experiment in Section 4.1 interesting, but not very conclusive. If you want to show that the proposed approach focuses more on the actual object instead of the background, then the proposed method should be evaluated subject to shortcut learning.
> > Below I give a more detailed list of the issues I perceive in the present revision.
> >
> > ## Details
> > * The definition of $X^{(n)}$ is ambiguous. In Def 3.1 you state it's the output of the neural network, so I'd assume that it's the output of the final softmax function, but that's not true, it's the last feature map or I guess more correct would be that it's the output of the penultimate layer/ last convolutional layer.
> > * The definition of the ERF as $\frac{X_{ij}^{(n)}}{X_{ij}^{(0)}}$ is wrong, the pixels don't have to be and also can't in general be the same because input and the last convolutional layer don't necessarily have the same dimensionalities.
> > * The use of the term pixel for input and feature map elements is highly confusing. Many times the reader has to infer what is meant from context, which is often not sufficiently defined.
> > * Why do you assume each layer has only a single channel for the definition of the ERF? Just state how this definition can be generalized to multiple layers.
> > * In Def. 3.2 $y$ is used as a variable for the feature map, later in Eq. (1) $y$ describes the labels. Why do you redefine $y = X^{(n)}$ instead of using directly $X^{(n)}$? Things like that make it really hard to follow.
> > * Def 3.4: $k$ is the channel index, not the number of channels
> > * Eq. (1) $X$ should be $X^{(n)}$?
> > * Section 4.1: "Each pixel in the input image corresponds to to a pixel in the output feature map" -> How do you make that statement? That's not even true after the first convolution.
> > * Why does the ERF distance take the absolute value of ERF(0,0)-ERF(w/2,h/2)? This way larger is not necessarily better (as stated in Tbl 2), because it could be that ERF(0,0) is much higher than ERF(w/2,h/2).
> > * "Table 2 shows that for the standard CNN each pixel has the same effect on the output" -> You can impossibly draw that conclusion from the experiment that only compares one corner against the center.
> > * Section 4.3: "destroying the gradient" is not a well-defined term. Why are only the clean accuracies shown in Table 5, when it's supposed to compare against robust competitors? The competitors should be evaluated just as in Table 3.
> > * Adv. training on Cifar-10 takes a bit, but it's absolutely doable even on colab. It's still unclear on which attack SAT is trained and which norm is used for PGD.

---

> > > ### Author Response · Authors · 2024-06-03
> > >
> > > Thank you for taking the time to read the revision in such detail. We updated the definitions and text as requested:
> > >
> > > - Updated definition 3.1: $X^{(n)}$ is feature output, changed coordinates in the feature layers
> > > - Replaced "pixels" by "location"/"positions" for the feature output. Only for the input, we use "pixels" now.
> > > - Explained the case of multiple output channels
> > > - Def. 3.2 use directly $X^{(n)}$ instead of $y$
> > > - Emphasized that with our modified model the last feature map is at $X^{(n+1)}$ (no longer $X^{(n)}$)
> > > - Updated definition of ERF distance
> > > - Specified use of \ell_{\infty} for PGD/AT
> > > - Replaced "destroying the gradient" by "shattered gradients" [1]
> > > - More text improvements
> > >
> > > The changed text was highlighted in blue. We are happy to make any further changes or conduct additional experiments as needed.
> > >
> > > Furthermore, we intend to make our code publicly available. This ensures reproducibility.
> > >
> > > We want to explain some aspects in more detail and highlight our contributions.
> > >
> > > ## Additional visual experiment
> > >
> > > We have added a new experiment that visualizes the effect of the ERF for 9 different positions. It can be found in B.2. (appendix). It qualitatively confirms our hypothesis that certain input pixels become less important for classification.
> > >
> > > In the previous revision, we wrote “Each pixel in the input image corresponds to a pixel in the output feature map”. We agree that this was not precise. We were referring to the fact that the location is approximately the same (ERF). For example, the ERF(1, 7) corresponds to the top right corner in the original input image (see Fig. A1/A2 appendix). We have clarified this in the text.
> > >
> > > As for “shortcut learning”, we can already see from the adversarial attacks, ERF, distributional shift accuracy and visual experiments that the network focuses less on the background. The background can still be a feature, but to a lesser extent than in a standard network. Further work can attempt to improve this even more.
> > >
> > > ## Other non-AT approaches
> > >
> > > Table 5 highlights that other approaches that achieve robustness against adversarial attacks, show significant decreases in accuracy. However, then these methods do not hold any advantage with respect to AT. The main argument in favor of non-AT methods is that they are just as fast as standard networks and have no loss of accuracy.
> > >
> > > However, FLC Pooling uses 2x 2D FFTs which have a high time complexity. It also sees a drop of 16% accuracy on CUB.
> > >
> > > Similarly, we found more than 20% reduction in accuracy for k-WTA.
> > >
> > > In both cases, we could just as well use adversarial training, as this also leads to a loss of accuracy, while at the same time achieving a significantly higher robustness.
> > >
> > > We therefore omitted the robustness experiment, as the accuracy of the two methods is too low.
> > >
> > > We performed some quick experiments. For CUB, we achieve 4 % FGSM robustness with k-WTA. Our approach achieves 21.2 % FGSM robustness. In addition, the original paper reports a robustness of 13.3 % for PGD on CIFAR-10. We achieve 51.7 %.
> > >
> > > ## Sparsity loss
> > >
> > > In the new experiment in the appendix "section F", we show that simple \ell_1 regularization cannot be used to achieve the same results. The decrease in accuracy would be too high, and it is unable to control the number of pixels. Refer to 5Kpo "Robustness".
> > >
> > > [1] https://arxiv.org/pdf/1802.00420

---

### Review · Reviewer_uKRt · 2024-05-11

**Summary Of Contributions:**

- This paper proposes the method (Top-GAP) for achieving adversarial robustness by combining global average pooling (GAP) and a feature pyramid network (FPN), without the need for generating adversarial examples that require high computational resources.

- It achieves robust accuracies of over 50% against FGSM/PGD.

**Audience:**

Yes

**Broader Impact Concerns:**

It is well known that standard accuracy and robust accuracy are generally in a trade-off relationship [3, 4]. However, the authors claim that clean accuracy and robust accuracy are highly correlated when using the proposed method. I believe that sufficient experiments and discussions need to be conducted to support this claim, including comprehensive validation of the methodology using reliable attacks such as AA [2].

[2] Croce et al. Reliable evaluation of adversarial robustness with an ensemble of diverse parameter-free attacks. https://arxiv.org/abs/2003.01690.
[3] Zhang et al. Theoretically Principled Trade-off between Robustness and Accuracy. https://arxiv.org/abs/1901.08573.
[4] Mady et al. Towards Deep Learning Models Resistant to Adversarial Attacks. https://arxiv.org/abs/1706.06083.

**Claims And Evidence:**

No

**Requested Changes:**

1. Notational Ambiguity: Overall, the mathematical formulations are not precisely written, making it difficult to understand.

- In equation (1), clarification on the indices {i, j, t} would be helpful.
- In equation (2), clarification on the indices {i, t} would be beneficial.
- In equation (3), it would be beneficial to explicitly specify the learning parameters within the loss function.

2. In equation (3), it was stated that \lambda=1 is sufficient; however, including an ablation study with \lambda=0, along with the clear demonstration of the effect of the L1 penalty term and its sensitivity, would be beneficial.

3. The precise meaning of "dropout deactivated" during the prediction phase refers to whether dropout layers are active or inactive during inference. Generally, during evaluation mode, dropout is deactivated, and the dropout rate is multiplied during inference. Could you clarify if you're asking for an explanation of this convention or if you're referring to inference without multiplying the dropout rate?

4. During inference or when generating outputs such as CAM, dropout is not used. Why is dropout used during the training phase then? Is dropout itself one of the key components of this paper?

**Strengths And Weaknesses:**

Strengths

- The authors propose a method to train adversrially robust model without generating computationally expensive adversarial examples.

- The authors argue that their proposed methodology does not exhibit a trade-off relationship between standard accuracy and robust accuracy. To my best knowledge, this perspective could offer a new insight into the adversarial robustness community. However, it needs to be thoroughly examined through extensive experiments.

Weaknesses

1. The significant difference in rob_acc between PGD20 and PGD50 (Table 4) suggests the occurrence of gradient masking [1]. Verification of gradient masking is crucial in the adversarial robustness community. I recommend referring to [1] or [2] for confirmation.

2. Observing Figure 5, it seems that the impact of the L1 loss outweighs that of global average pooling. Nonetheless, this may not be the primary focus of the paper.

[1] Carlini et al. On Evaluating Adversarial Robustness. https://arxiv.org/abs/1902.06705.
[2] Croce et al. Reliable evaluation of adversarial robustness with an ensemble of diverse parameter-free attacks. https://arxiv.org/abs/2003.01690.

---

> ### Author Response · Authors · 2024-05-24
>
> Thank you for the feedback, we have made various changes to clarify the notation and added new experiments. Refer to the general comment above for the list of all changes.
>
> **Gradient masking**
>
> Using stronger attacks like AutoAttack reduces our robustness. However, we would like to emphasize that we get our robustness "for free". Accuracy does not decrease, while many other methods can lead to a sharp drop in accuracy, more GPU resources and/or convergence issues. In practice, this means that our method can be used without noticing any negative effects.
>
> So although we do not achieve high robustness against all attacks, we are still robust against some of the attacks in AutoAttack. Standard AutoAttack uses the following attacks: square, apgd-ce, apgd-t, fab-t. The results on CIFAR-10 are as follows:
>
> - square: 34%
> - apgd-ce: 7% (20 iterations)
> - apgd-t/fab-t: 0% (20 iterations)
>
> We have also added this experiment to the text. Notably, square attack does not rely on gradient information but we achieve good results.
>
> **Figure 5 (now Figure 6)**
>
> Using l1 loss alone also reduces the number of pixels. However, this loss function is too strong for certain datasets. For ImageNet, we observed a drop in accuracy of about 10% when only L1 loss was used. For other datasets, the decrease may not be as severe (e.g. COCO). But more importantly, by using top-k pooling, we can control the number of pixels. Therefore, apart from leading to higher accuracy, it is needed for our "size prior" method. The ablation study in Table 7 shows that the combination of all three components lead to the best results.
>
> **Notation**
>
> We have improved the notation and added new terminology.
>
> **\lambda**
>
> In Table 8, we can see the results for \lambda=0. This corresponds to not using L1 regularization. Using different values, such as \lambda=0.5, only lead to differences of around 0.1\% in accuracy. Training for every architecture and every dataset models on different \lambda would be too time-consuming.
>
> **Dropout**
>
> Dropout is not a key component, but we have observed slight improvements in accuracy when using dropout. Our use of dropout is no different from its use in normal neural networks. We just wanted to show what exactly happens during training and prediction.
>
> We agree that this part was confusingly written, and have improved the text by removing non-essential text.
>
> **Clean accuracy and robust accuracy**
>
> We have partially reduced our claims in the text. We emphasize that we achieve increased robustness, but not SOTA robustness. However, this is not a drawback, as we get the robustness while keeping the same accuracy and speed. Our method can also be seen as a regularization that improves certain properties of the neural network (different effective receptive field, increase in IOU, improvements with respect to distribution shift).

---

> > ### Comment · Reviewer_uKRt · 2024-06-10
> >
> > Thank you for your response. However, there are still concerns regarding the main issues. The authors claim that adversarial robustness can be achieved for free using the proposed methodology, but I partially disagree with this. Methods that induce gradient masking to provide defense are currently considered failed defense strategies within the adversarial robustness community [1]. As the authors claim that their method becomes robust against specific attacks, it might be significant only if it shows superior performance compared to other methods against certain non-gradient-based defenses such as SQUARE. However, in its current version, it is difficult to confirm this.
> >
> > [1] Obfuscated Gradients Give a False Sense of Security: Circumventing Defenses to Adversarial Examples. ICML. 2018.

---

> > > ### Author Response · Authors · 2024-06-10
> > >
> > > Thank you for your comment. We are aware that Square would be a good test to show that we do not have gradient masking. For this reason, we have included an experiment against this adversarial attack in the current revision. We show **34% robustness against Square** on CIFAR-10. This experiment can be found in the current revision on page 10 of the document.
> > >
> > > Apart from the Square attack, another proof that we do not have gradient masking is our accuracy. For example, k-WTA activation [1] achieves robustness by using a discontinuous function that invalidates the gradient of the model. For this reason, k-WTA achieves 30% lower accuracy in our experiments (Table 5). It is more difficult to train the network.
> > >
> > > Our model converges to the same high accuracy as a standard model. This shows that we achieve "true" robustness. Moreover, we also have higher robustness to distribution shifts.
> > >
> > > [1] Enhancing Adversarial Defense by k-Winners-Take-All https://arxiv.org/pdf/1905.10510

---

### Author Response · Authors · 2024-05-24
**Summary of changes**

First of all, we would like to thank all reviewers for their insightful comments and concerns. All reviewers appreciated that our approach provides robustness without adversarial training. They also found the idea of limiting the number of pixels interesting.

Nevertheless, there were doubts and criticisms, which we hope to have addressed sufficiently with our major changes to the paper. We have marked all changes in red in the original document. We will be happy to make further changes and respond to further questions.

Above all, we would like to point out that AutoAttack is a fairly powerful attack, but at the same time expensive and easy to detect. With adversarial training, we would achieve robustness against this attack. However, the trade-off is a degradation in accuracy. Our goal is to regularize the network in such a way that we get both high clean accuracy and robust accuracy on real-world datasets. For this reason, we did not limit ourselves to ImageNet or CIFAR only. We can show that other non-AT approaches have weak clean accuracy on real-world datasets (new experiment).

Finally, this regularization also shows that there are significant changes in the way the network works. In a new experiment, we look at the effective receptive field to characterize< which pixels have an impact on the output.

## List of changes to the paper

- Show that we do not have gradient masking by applying square attack. Square attack does not use gradients.
- Toned down the claims in the paper with respect to adaptive adversarial attacks.
- Added an experiment showing higher accuracy than other non-AT techniques
- Restructured the evaluation section by focusing on specific hypothesis
- Moved several experiments to the appendix to focus more on robustness and interpretability
- Added an experiment to show that our network uses fewer pixels for classification
- Rewrote the method section to define several terms more concretely.
- Added definition for the effective receptive field

---

### Decision · Action_Editor_2HA8 · 2024-06-17

**Recommendation:** Reject

**Comment:**

The recommendation is based on the reviewers' comments, the action editor's evaluation, and the authors’ response.

This paper proposed a regularization method (Top-GAP) to increase the empirical robustness of CNNs against some white-box/black-box attacks, as well as distributional shifts. While the authors' rebuttal has addressed some of the reviewers' concerns, the majority of reviewers share common concerns about the technical contributions. Therefore, this submission should not be accepted in its current form. Specifically, the remaining issues and suggestions are listed below:

- It is necessary to thoroughly assess gradient masking across a wide range of datasets to ensure definitive resolution.
- A literature review and comparison with methodologies that achieve adversarial robustness without generating adversarial examples are essential.
- Experiments indicate that the proposed method relies less on background information, but unfortunately, there is no systematic experiment design that would make a strong argument in this direction. There is no comparison to other methods to reduce the information used in the background for example.
- The focus on adversarial robustness is not well substantiated. The comparison against other related robustness methods (Table 5) is not very convincing, it is conducted on only one dataset and only clean accuracy is evaluated.
- The robustness drops to 0% when employing a strong attack such as AutoAttack. This questions the practicality of the proposed method.
- Why GradCAM-based regularization can improve adversarial robustness needs further (and more direct) justification. For example, will the certified radius be enlarged using the proposed method?

 I hope the reviewers’ comments can help the authors prepare a better version of this submission.

**Audience:**

Yes. The topic is of interest to the general audience.

**Claims And Evidence:**

The claims and empirical results are limited in scope. Please see "Comments" for details.

**Resubmission Of Major Revision:**

The authors may consider submitting a major revision at a later time.

---

> ### Author Response · Authors · 2024-06-18
> **Thanks to the reviewers and editor**
>
> We would like to thank the reviewers and editor for their efforts and detailed feedback. Despite the outcome, we very much value quality of this review process and will rebuild this paper from scratch based on the suggestions made.